# Bacterial Volatiles Known to Inhibit *Phytophthora infestans* Are Emitted on Potato Leaves by *Pseudomonas* Strains

**DOI:** 10.3390/microorganisms10081510

**Published:** 2022-07-26

**Authors:** Aurélie Gfeller, Pascal Fuchsmann, Mout De Vrieze, Katia Gindro, Laure Weisskopf

**Affiliations:** 1Changins School of Viticulture and Oenology, 1260 Nyon, Switzerland; aurelie.gfeller@agroscope.admin.ch (A.G.); mout.devrieze@unifr.ch (M.D.V.); 2Agroscope, Plant Protection, 1260 Nyon, Switzerland; katia.gindro@agroscope.admin.ch; 3Agroscope, Nutrition, Sensory analysis and Flavour Group, 3003 Bern, Switzerland; pascal.fuchsmann@agroscope.admin.ch; 4Department of Biology, University of Fribourg, 1700 Fribourg, Switzerland

**Keywords:** bacterial volatile emission, phyllosphere, 1-undecene, dimethyl disulfide, dimethyl trisulfide, *Solanum tuberosum*

## Abstract

Bacterial volatiles play important roles in mediating beneficial interactions between plants and their associated microbiota. Despite their relevance, bacterial volatiles are mostly studied under laboratory conditions, although these strongly differ from the natural environment bacteria encounter when colonizing plant roots or shoots. In this work, we ask the question whether plant-associated bacteria also emit bioactive volatiles when growing on plant leaves rather than on artificial media. Using four potato-associated *Pseudomonas*, we demonstrate that potato leaves offer sufficient nutrients for the four strains to grow and emit volatiles, among which 1-undecene and Sulfur compounds have previously demonstrated the ability to inhibit the development of the oomycete *Phytophthora infestans*, the causative agent of potato late blight. Our results bring the proof of concept that bacterial volatiles with known plant health-promoting properties can be emitted on the surface of leaves and warrant further studies to test the bacterial emission of bioactive volatiles in greenhouse and field-grown plants.

## 1. Introduction

Plants are densely colonized from the roots to the shoots by a plethora of microbes and increasing evidence indicates that these microbiota play essential roles for the growth, development and defense of its plant host [1,2,3,4]. Bacteria belonging to the genus *Pseudomonas* are an important component of this diverse plant microbiota and have been shown to display many different plant-beneficial properties, and, in particular, to be very well equipped to help their host defend itself against disease-causing agents, e.g., through induction of systemic resistance, niche competition or the production of antimicrobial compoundsac [5,6]. Among the wide variety of chemical weapons encoded in the genomes of many plant-associated *Pseudomonas* strains, volatile organic compounds (VOCs) have recently attracted increased attention as short- and long-distance bioactive compounds with strong inhibiting effects against various disease-causing agents [7,8,9,10,11,12]. Although initially demonstrated on the model plant *Arabidopsis* [13,14], the bioactivity of these volatiles has meanwhile also been reported in many plants of agronomical relevance such as grapevines, peppers, tomatoes or potatoes [15,16,17,18]. This latter crop is mainly threatened by the oomycete *Phytophthora infestans* (Mont.) de Bary, the causative agent of late blight, which leads to high economic losses worldwide. Over 150 years after the Irish Famine partly caused by this pathogen, we are currently still struggling to control this disease and rely to this end on repeated applications of synthetic or copper-based fungicides with deleterious side-effects on environmental and human health [19,20,21]. 

In an attempt to find alternative solutions to control potato late blight, we have previously isolated and characterized many members of its microbiota [9]. From the assembled strain collection, *Pseudomonas* strains isolated from both the roots and the shoots were the emitters of the most bioactive volatiles, and we identified 1-Undecene as well as diverse Sulfur-containing volatiles (S-VOCs), which were particularly efficient in inhibiting different life stages of *P. infestans* [9,16,22]. Some of these S-VOCs were shown in other studies to contribute to plant Sulfur nutrition or to induced systemic resistance, highlighting their versatile nature, which could be of great interest for practical applications [23,24].

However, so far, all studies dealing with the identification and characterization of volatile emissions by plant-associated bacteria have been carried out in artificial laboratory media, which represent very different growth conditions from those encountered by bacteria on their plant host. This is especially true for the phyllosphere environment, which in contrast to the rhizosphere is very poor in the organic carbon sustaining the growth of heterotrophic bacteria such as *Pseudomonas.* Therefore, the aim of this study was to assess whether such bioactive volatiles would also be emitted by bacteria growing on leaves without any external supply of nutrients. 

Using leaves of in vitro grown potato plants inoculated or not with different *Pseudomonas* strains and taking advantage of a specific volatile analysis method allowing sensitive detection of S-VOCs, we provide evidence that potato leaf-colonizing bacteria emit detectable quantities of volatiles with known anti-*Phytophthora* activity. Although the system used in the present study lies still far away from the reality of field-grown potato plants, our results bring the important proof of concept that the emission of volatile compounds by bacteria is not a laboratory artefact but occurs also in the nutrient-poor leaf surface environment. 

## 2. Materials and Methods

### 2.1. Cultivation of Microbial Strains and of Plants

#### 2.1.1. Bacterial Strains

The four *Pseudomonas* strains used for this study were isolated in 2015 from field-grown potatoes as described in [9,25], two from the roots (*Pseudomonas donghuensis* R32, *P. chlororaphis* R47) and two from the shoots (*Pseudomonas* sp. S04, *Pseudomonas* sp. S35). We selected these strains because they differed in their protective activity against *P. infestans* and in their ability to emit VOCs on laboratory media. The strains were kept in 25% (*v*/*v*) glycerol for long-term storage and passaged weekly on 1.5% agar (*w*/*v*; Erne, Switzerland) LB medium (20 g/L; Applichem, Schaffhausen, Switzerland) or ABG medium [26] for routine use.

#### 2.1.2. *Phytophthora Infestans*

The isolate *P. infestans* was originally isolated by H. Krebs (Agroscope, Switzerland) in 2001. It was later characterized as an A2 mating type isolate belonging to the lineage Blue13 [27]. *P. infestans* was maintained on pea growth medium [28] or rye agar [9] on Petri dishes sealed with Parafilm (BEMIS Flexible Packaging, Neenah, WI, USA) and incubated at 19 °C in the dark. 

#### 2.1.3. Potato Plantsa

Sterile in vitro potato plantlets (var. Désirée) were obtained from E. Droz (Agroscope, Switzerland) and cultivated on CMS medium, a minimal medium containing per liter 20 g sucrose, 8 g Agar-Agar (Kobe), 230 mg NH_4_H_2_PO_4_, 370 mg MgSO_4_ • 7 H_2_O, 1.21 g KNO_3_, 708 mg Ca(NO_3_)_2_ • 4 H_2_O, 37 mg Na_2_EDTA, 9.3 mg FeSO_4_ • 7 H_2_O, 8.15 mg KI, and 1 mL CMS solution (containing per liter 11.1 g MnSO_4_ • 4 H_2_O, 8.41 g MnSO_4_ • H_2_O, 2.87 g ZnSO_4_ • 7 H_2_O, 0.25 g CuSO_4_ • 5 H_2_O, 0.25 g Na_2_MoO_4_ • 2 H_2_O and 25 mg CoCl_2_ • 6 H_2_O). For propagation, one internode with an intact leaf was taken and placed into a new culture tube containing 20 mL of CMS medium. Plants were grown for 30 d in a culture room (16H/8H D/N; 20 °C). After 30 d, the upper part of the plant with the four last expanded leaves was collected for the inoculation and volatile analysis.

### 2.2. Inoculation of Plants and Collection of Volatiles

Sterile 20 mL GC headspace screw vials were filled with 4 mL of water agar (WA). To prepare the bacterial culture, four colonies of an overnight culture (24 °C) of each strain were resuspended in 500 µL 0.9% NaCl (*w*/*v*) and 100 µL were plated on ABG medium. Plates were incubated overnight at 24 °C. The day after, the bacterial lawn was resuspended in 10 mL 0.9% NaCl (*w*/*v*), washed two times with 0.9% NaCl (*w*/*v*) and one additional time with sterile deionised water. After these three washing steps aiming to remove traces of the cultivation medium, bacteria were finally resuspended in sterile deionised water. Sporangia of a 3-week-old P. infestans culture were harvested with sterile deionised water. 

Potato leaves were dipped in (i) sterile water (control), (ii) washed bacterial solution at OD_600_ = 1 of each strain separately, (iii) a sporangial solution of 125,000 sporangia/mL, or (iv) a mixed solution of bacteria (OD_600_ = 1) and sporangia (125,000/mL). After dipping, leaves were carefully deposited on the top of the WA in the screw vials, caps were applied and screwed by letting one quarter turn open to allow for air penetration. Vials were kept in the dark at 20 °C for two days, after which they were completely closed and left for two additional days at 20 °C in the dark until volatile collection (see below). The biomass of each plant sample was determined by weighing the vial before and after addition of the leaves. 

### 2.3. Analysis of the Emitted Volatiles by HS-SPME GC-PFPD/MS 

Samples were extracted using a 2 cm solid phase microextraction (SPME) carboxen/polydimethylsiloxane (CAR/PDMS) 85 μm StableFlex fiber (Supelco, Bellefonte, PA, USA). The fiber was conditioned according to the supplier’s recommendations (300 °C for 60 min under a nitrogen flow in a conditioner SPME station). The analyses were realized using an MPS2 autosampler equipped with Maestro1 software, V.1.4.8.14/3.5 (Gerstel, Sursee, Switzerland), a Trace GC Ultra GC coupled with a DSQ II mass selective detector (MSD) (Thermo Finnigan, Milan, Italy), and a pulsed flame photometric detector (PFPD) (OI Analytical, College Station, TX, USA) as a specific detector for sulfur compounds, as described in [29]. 

The headspace was extracted for 120 min at 60 °C with an agitation rate of 250 rpm without incubation. Bound volatiles were desorbed for 1 min at 250 °C in the injector, which was in the splitless mode for 30 s, and then the split valve was opened (split flow = 80 mL/min). VOCs were separated on a TRB-FFAP fused silica capillary column (100% polyethylene glycol PEG with nitroterephthalic acid, bonded and cross-linked, 30 m × 0.32 mm, 1.0 μm film, Teknokroma, Barcelona, Spain) with helium as the carrier gas at a constant flow of 2.1 mL/min (37 cm/s). The oven temperature was programmed as follows: 4 min at 40 °C, then heated to 220 °C at a rate of 5 °C/min, with a final hold time of 5 min. The settings of the PFPD were as follows: 250 °C, voltage at 540 V, ignitor current at 2.6 A, trigger level at 300, range at 10, and attenuation at 32 with flow rates of air1 at 17 mL/min, H_2_ at 14 mL/min, and air2 at 10 mL/min. The MS settings were as follows: transfer line at 230 °C, source temperature at 230 °C, and analytes monitored in SCAN mode between 30 and 150 amu without solvent delay. 

The detector response signals were integrated using the Xcalibur 2.0.7 software (Thermo Fisher Scientific AG, Reinach, Switzerland). The NIST/EPA/NIH mass spectral library (NIST11) version 2.0 g (NIST, Gaithersburg, MD, USA) was used for peak identification. The PFPD detector was configured to detect and trace only the volatile organic sulfur compounds (S-VOCs) in the chromatogram. Quantitative ion peak area was integrated in order to evaluate the abundance of each VOC. This amount was then divided by the fresh weight of the leaves contained in each vial to estimate the volatile emission per unit of biological material and allow comparisons between the different samples. 

### 2.4. Data and Statistical Analysis

The leaf inoculation and volatile collection experiment was performed in three independent biological replicates containing each two technical repetitions. In each independent experiment, the mean of the two technical repetitions was calculated for each treatment. ANOVA of the VOCs quantitative ion peak area/g in conjunction with Tukey’s test (unadjusted *p* < 0.05) was applied using the software package R (2.8.0). Data were centered and assigned to different qualitative groups (R32, R47, S04, S35, non-inoculated). PCA was performed using the ‘FactoMineR’ function of the R-package.

## 3. Results 

### 3.1. Inoculation with Pseudomonas strains Leads to Major Changes in Potato Leaf Volatile Emissions

In order to characterize the volatile blends emitted by potato-associated *Pseudomonas* when growing on their host plant rather than on artificial laboratory media, four selected *Pseudomonas* strains (R32, R47, S04 and S35) were inoculated on sterile potato leaves. Since we were also interested to see whether concomitant inoculation and infection with *Phytophthora infestans* would alter the volatile profiles of the strains, we performed the analysis on leaves inoculated with the bacterial strains alone or together with *P. infestans.* The first question was whether the inoculated strains would be able to survive and grow on sterile potato leaves. Our measurement of bacterial density a few hours and five days after inoculation revealed that the four strains were able not only to survive, but to grow well on the leaves, reaching ca. 10^11^ cells per leaf (Appendix A), which was a prerequisite for their emission of VOCs. To analyse the VOCs they emitted, we used a combination of two detection methods: PFPD to identify Sulfur-containing volatile organic compounds (S-VOCs) with increased selectivity, and mass spectrometry (MS) to identity all detected compounds. A total of 75 VOCs were detected in the samples including 13 S-VOCs detected specifically with the PFPD detector. 

An ANOVA analysis on these 75 detected VOCs revealed 32 VOCs which were differentially emitted by the leaves depending on their inoculation status (differing either between non-inoculated and inoculated leaves or between leaves inoculated with different strains, see Appendix A for details). In contrast, only five VOCs were emitted differentially in leaves infected vs. non-infected with *P. infestans* (Table 1). 

A principal component analysis (PCA) was then performed on these 32 differentially emitted VOCs to identify the most discriminative factors between the different samples. As expected from the results shown in Table 1, PCA did not discriminate between leaves inoculated with *P*. *infestans* and uninoculated leaves (Appendix A). For this reason, PCA is presented with only the factor strain (Figure 1). The main principal component (PC1) accounted for 38% of the variance, while PC2 accounted for 16% of the variation.

A clear difference could be observed in the VOCs emitted by leaves inoculated with *Pseudomonas* sp. S35 on PC1 (Figure 1A), while PC2 discriminated non-inoculated leaves and those inoculated with S04, R32 and S35 from those inoculated with *P. chlororaphis* R47. Figure 1B shows the vectors of the 16 VOCs that were best represented on the projection of the two axis and therefore likely responsible for the differences in volatilome profiles. 

### 3.2. Volatiles with Known Anti-Phytophthora Activity Are Emitted on Potato Leaves by Pseudomonas sp. S35

We then compared the emission of the 16 discriminative compounds mentioned above in the different samples (leaves non-inoculated or inoculated with the four *Pseudomonas* strains) using Tukey post hoc comparison (*p*-value < 0.05) (Figure 2). 

This analysis revealed 11 VOCs to be emitted in higher quantities in leaves inoculated with *Pseudomonas* sp. S35, among which most S-VOCs and 1-Undecene. It is worth noting that many of these compounds were previously shown to inhibit *P. infestans* in in vitro experiments. Inoculation of the leaves with *P. chlororaphis* R47 also led to higher emission of four VOCs, while the other *Pseudomonas* strains only marginally affected the volatile profiles of the leaves.

## 4. Discussion 

The importance of volatile organic compounds (VOCs) for microbial communication and for the establishment of mutualistic interactions with other microbes, with plants and with animals is increasingly recognized [47,48]. Microbes, and bacteria in particular, are found in a wide range of environments differing in their physical and chemical properties, as well as in their nutrient availability. These factors are expected to exert a strong influence on the quality and quantity of the emitted VOCs. Accordingly, earlier studies have shown that similar strains can emit very different VOCs depending on their cultivation medium or growth phase [49,50,51,52]. The same laboratory studies have shown that even very nutrient-poor media are sufficient to support the emission of VOCs, triggering e.g., plant growth promotion [51] or inhibition of fungal phytopathogens [49]. Such inhibiting activity of bacterial volatiles on plant disease-causing agents is of great interest considering the need to find alternatives to the synthetic fungicides currently used to protect plants. However, the ability of plant-associated bacteria to emit such potent volatiles has been so-far only demonstrated in laboratory conditions, and the question remained whether they would also be emitted by bacteria growing on nutrient-poor leaf surfaces. 

Using four *Pseudomonas* strains previously isolated either from the roots or from the shoots of potato plants, we have shown in this work that such volatile emission can indeed also occur on plant leaves. One particular volatile was detected only in leaves harboring *Pseudomonas* bacteria, namely 1-undecene, which is a well-known volatile emitted by members of this genus, and is even used as marker of infection by the widespread pathogen *P. aeruginosa* [53]. Plants, in contrast, are not known to produce this alkene, which indicates that the VOCs detected in our study are indeed produced by the bacteria and not by the plant as a reaction to bacterial inoculation. Likewise, the S-VOCs detected in our study such as dimethyl disulfide or dimethyl trisulfide are not known to be emitted by potato, although they are produced by other plants belonging to the *Brassicaceae* or the *Alliaceae* families [54]. In addition to being a marker of the genus *Pseudomonas,* 1-undecene was shown to strongly inhibit zoospore release in the late blight-causing oomycete *Phytophthora infestans* [9]. The fact that potato-associated *Pseudomonas* are able to emit this volatile at the site of infection is therefore of particular relevance for their putative use as biocontrol agents, although we cannot warrant at this stage that 1-undecene or the S-VOCs would be emitted in sufficient concentrations to support anti-*Phytophthora* activity. Indeed, this study only allowed comparative analysis of the emitted VOCs, but did not allow to determine the absolute concentrations found in the leaf headspaces, which is a goal of future research.

One surprising observation we made was the low number of VOCs differentially emitted by leaves infected vs. non-infected with *P. infestans* in our experimental conditions. This was probably due to a very mild infection that could have been caused by low fitness of the sporangia used to inoculate the leaves, or to the relatively short time span of the incubation, which did not allow the infection to fully develop. It might also be that infection with a hemibiotrophic pathogen does not induce the same extent of changes in VOCs emissions as those reported after attacks by herbivores or necrotrophic fungi [55]. 

Interestingly, the *Pseudomonas* strains used in this study differed in their VOCs profiles when growing on leaves, as illustrated in Figure 2. From the four strains studied, two stood out, with higher divergence from the non-inoculated control leaves: *P. chlororaphis* R47 and *Pseudomonas* sp. S35. These strains differ in many aspects, including the environment they were isolated from (potato rhizosphere for R47 and phyllosphere for S35) or their genome size and content (R47 harboring a wide variety of genes encoding antimicrobial compounds and S35 harboring several genes involved in plant colonization) [25]. While R47 showed very strong in vitro inhibition of *P. infestans* at various developmental stages, S35, which was largely inactive in vitro, provided the most consistent in planta protection against late blight in three different potato cultivars [25,56]. Although infection assays usually do not allow to separate VOCs-mediated effects from direct effects, our present study showing the emission of a range of anti-*Phytophthora* volatiles (e.g., 1-undecene, heptanone, dimethyl disulfide, and dimethyl trisulfide [22]) from the surface of S35-inoculated leaves highlights these VOCs as candidate compounds underlying at least partly the very good in planta protection conferred by this strain against *P. infestans* infection.

In conclusion, our study brings the proof of concept that bacterial VOCs can be emitted on leaf surfaces and not only on laboratory media. This further strengthens the translational potential of these bioactive metabolites and of their emitting strains, which could be used either as inducers of systemic resistance or as direct inhibitors of the pathogens’ growth and development. However, both the metabolites and the strains producing them would then need to withstand a yet harsher environment than that encountered on our in vitro potato leaves, facing e.g., UV light or drastic changes in temperature and humidity. Further studies shall therefore investigate the true potential of such in situ bacterial VOCs emission on plants growing under greenhouse and field conditions. 

## Figures and Tables

**Figure 1 microorganisms-10-01510-f001:**
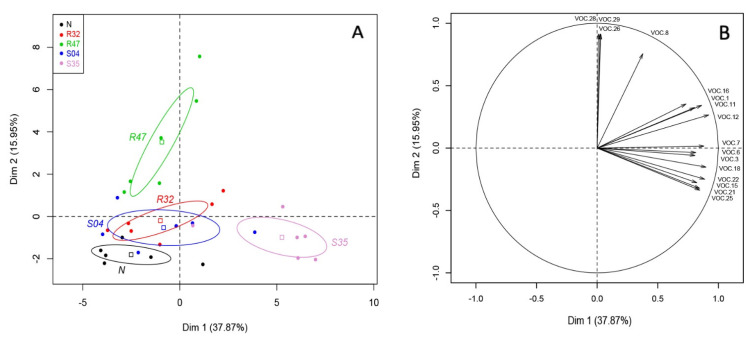
Principal component analysis of the volatiles emitted by leaves inoculated with *Pseudomonas* strains. (**A**) Score Plot of PC1 and PC2 of potato leaves non-inoculated (N) or inoculated with different *Pseudomonas* strains (R32, R47, S04, S35) following PCA. Superimposed on the plot are mean scores on the components for qualitative variables that are included in the PCA command and the colored ellipses illustrating the 95% confidence interval with each score. (**B**) Results of principal components analysis (PCA): projection on two first factors (PC1 and PC2) of the VOCs with a cosinus square superior to 0.6. VOC.1: 2-methyl-3-methylthiofuran; VOC.3: dimethyl disulfide, VOC.6: dimethyl trisulfide, VOC.7: methanethiol, VOC.8: thiocyanic methylester, VOC.11: 3-pentanone, VOC.12: Unknown 5C, VOC.15: hexanal, VOC.16: 1-undecene, VOC.18: 2-heptanone, VOC.21: (Z)-2-heptenal, VOC.22: 6-octene-2-one, VOC.25: (E)-2-octenal, VOC.26: putative, longiverbenone or isomer, VOC.28: Cyclobutane, tetrakis (1-methylethylidene)-, VOC.29: (2,6,6-trimethylcyclohexen-1-yl) methylsulfonylbenzene.

**Figure 2 microorganisms-10-01510-f002:**
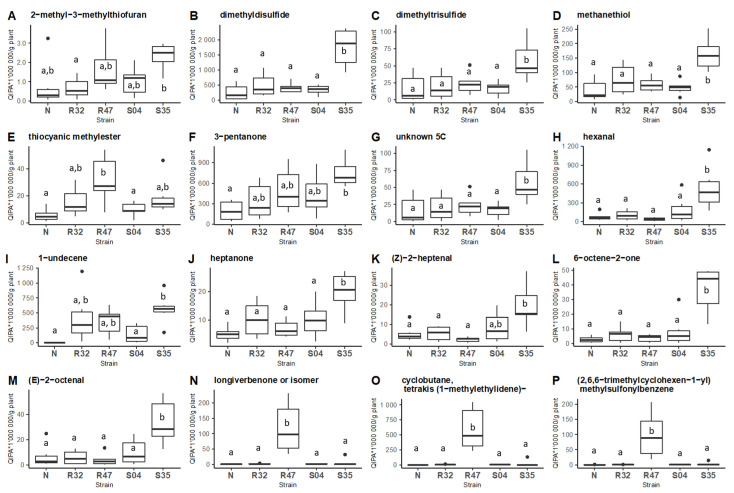
Comparative emission of VOCs by different *Pseudomonas* inoculated on potato leaves. Boxplot of the 16 VOCs with a cosinus square superior to 0.6 according to the two first dimensions of the principal component analysis (**A**–**P**). Tukey HSD post-hoc tests with Holm correction were performed on each VOC between potato leaves non-inoculated (N) or inoculated with different *Pseudomonas* strains (R32, R47, S04, S35; N = 6; Quantitative Ion Peak Area/g plant (QIPA/g plant)). Different letters indicate statistically significant differences at a significance level of *p* < 0.05).

**Table 1 microorganisms-10-01510-t001:** Differentially emitted VOCs in samples inoculated with bacterial strains *P. donghuensis* R32, *P. chlororaphis* R47, *Pseudomonas* sp. S04, *Pseudomonas* sp. S35 in presence and absence of *P. infestans* (n = 3). Only differentially expressed VOCs according to bacterial strain inoculation or infection status with *P. infestans* are presented.

CAS (1)	Name	Factor Strain (2)	Factor Pathogen (3)	Strain: Pathogen (4)	Qualifier and Quantifier Ions (5)	Identification (6)	Sample RI (7)	Ref RI (8)
74-93-1	methanethiol	***			45, **47**, 48	MS; STD; RI	799	800 [30]
75-18-3	dimethylsulfide	**			45, 47, **62**	MS; STD; RI	822	844 [31]
67-63-0	isopropyl alcohol	**			45, **59**	MS; RI	945	950 [32]
96-22-0	3-pentanone	*			29, 57, **86**	MS; RI	1001	970 [33]
	unknown-5C	***				MS	1003	
1629-58-9	1-penten-3-one	*			27, 55, **84**	MS; RI	1048	1034 [30]
71-23-8	1-propanol	**			31, 42, **59**	MS	1062	
624-92-0	dimethyldisulfide	***			45, 79, **94**	MS; STD; RI	1106	1057 [34]
66-25-1	hexanal	*	***	*	44, 56, **72**	MS; RI	1111	1082 [30]
821-95-4	1-undecene	**			43, 55, **70**	MS; STD; RI	1138	1135 [35]
625-33-2	3-penten-2-one		*		41, 69, **84**	MS; RI	1164	1123 [36]
123-35-3	beta-myrcene		*		41, 69, **93**	MS; STD; RI	1169	1161 [37]
110-43-0	2-heptanone	***	*		43, 58, **71**	MS	1208	
123-51-3	3-methyl-1-butanol	*			55, 57, **70**	MS	1237	
556-64-9	thiocyanic methylester	**			45, 58, **73**	MS; RI	1311	1276 [38]
1576-95-0	(Z)-2-penten-1-ol	*			57, **68**, 86	MS	1340	
57266-86-1	(Z)-2-heptenal	***			70, **83**, 112	MS	1356	
35194-31-1	6-octen-2-one	***			68, **108**, 126	MS	1359	
63012-97-5	2-methyl-3-methylthiofuran	*			99, 113, **128**	MS	1372	
38401-84-2	1,6-dioxaspiro[4,4]nonane, 2-ethyl	*			87, 98, **127**	MS; RI	1376	1353 [39]
3658-80-8	dimethyltrisulfide	**			79, 111, **126**	MS; STD; RI	1415	1381 [40]
3228-02-2/89-83-8	3-methyl-4-isopropylphenol ^t^ or p- thymol ^t^		*		91, **135**, 150	MS	1441	2196 [41]
2548-87-0	(E)-2-octenal	*			55, 70, **83**	MS	1457	
64180-68-3	Longiverbenone ^t^ or isomer	***			135, **148**, 218	MS	1468	2207 [42]
868-84-8	carbonodithioic acid, S,S-dimethyl ester		*		75, 94, **122**	MS	1484	1059 [43]
98-01-1	furfural	*			67, 95, **96**	MS; RI	1498	1457 [44]
88919-66-8	cyclobutane, tetrakis (1-methylethylidene)-	***			173, 201, **216**	MS; RI	1545	1522 ^n^
56691-74-8	(2,6,6-trimethylcyclohexen-1-yl)methylsulfonylbenzene	***			81, 95, **137**	MS	1580	
67-68-5	dimethylsulfoxide	***			45, 63, **78**	MS; STD; RI	1618	1553 [45]
1679-49-8	2(3H)-furanone, dihydro-4-methyl-	***			42, **56**, 100	MS	1663	
1449-49-6	cyclobutanone,2,3,3-trimethyl-	***			41, 55, **70**	MS	1778	
124-25-4	tetradecanal	*			**82**, 96, 168	MS; RI	1941	1930 [46]

(1) CAS number of compounds listed in order of elution from a TRB-FFAP fused silica capillary column. Source CAS: Scifinder^®^ (Chemical Abstracts Service, Colombus, OH, USA); Confirmation of the compound identity was done by comparing the three predominant ions with the Nist11 library databases (24 VOCs), the library retention index (17 VOCs) and standards (7 VOCs). Abundance of VOCs were tested for effects of ‘strain’ (2), ‘pathogen’ (3) and interaction ‘strain’ x ‘pathogen’ (4) in a two-way analysis of variance (ANOVA). Significant differences are marked with * *p*-value < 0.05; ** *p*-value < 0.01; *** *p*-value < 0.001. (5) Quantitative ion (in bold) and qualitative ions (6) Identification methods: MS, comparison of mass spectra with those of the Nist11 library; STD, comparison of retention time and mass spectra of available standards; (7) Retention indices on TRB-FFAP column, experimentally determined using a saturated n-alkane standard solution C9-C20; (8) RI, comparison of retention indices with those reported in the literature (see respective references in brackets) or ^n^, from NIST Spectra mainlib_153401; ^t^ Tentatively identified.

## Data Availability

Not applicable.

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
