# Peer review of "Bacterial Volatiles Known to Inhibit *Phytophthora infestans* Are Emitted on Potato Leaves by *Pseudomonas* Strains"

_microorganisms, 2022, doi:10.3390/microorganisms10081510_

Round 1

Reviewer 1 Report

This manuscript provides a brief report on the production of bacterial volatile organic compounds (VOCs) produced by plant-beneficial Pseudomonas spp. inoculated on potato leaves. The authors showed that four Pseudomonas strains inoculated on potato leaves (a relatively nutrient-poor environment) were capable to maintain/reach a somewhat high-population size 120h after inoculation (10^11 per leaves). In addition, the colonization of the potato leaves by plant-beneficial Pseudomonas spp. led to the emission of numerous VOCs, some of which were known to display anti-Phytophthora activity. I like the manuscript and I find it to be beautifully written, even though one important passage would benefit to be clearer. It fits well with a brief report manuscript, and I believe the findings of this manuscript to be noteworthy. Here are my comments that could help improving the manuscript :

L47-48 : Attributing the Great Hunger of 1845-1849 to P. infestans in a cause-and-effect relationship is very far from the truth, considering the high number of causes and/or aggravating factors (overpopulation, government failure, mass eviction, extreme poverty etc.). There are a lot of research on this traumatic, complex and polarizing event in the Irish history, and it is important to remember that numerous shipments of grain and other crops were exported from Ireland under military escort at the peak of the famine. I understand why plant pathologists and students like to use this example (one of the best examples of the damage a plant pathogen can cause if left unchecked) and there are not a lot of examples of plant diseases with this level of destruction (the other example that comes to my mind is the thousands of hectares of orchards devastated in the US by the fire blight disease). I would rewrite this sentence to present P. infestans as ‘’A’’ cause of the Irish Famine (and not ‘’THE’’ cause as it is implicitly stated in the current wording) – it could be just by adding ‘’partly’’ before ‘’caused’’.

L84-85 : Three out of the four bacterial strains used in this study are wrongly identified at the species level in the manuscript. You can use the Type (Strain) Genome Server (https://tygs.dsmz.de/) to check by yourself. Only R47 is correctly identified as a P. chlororaphis strain. R32 belongs to the species P. donghuensis (digital DDH of 95.5% with the type strain HYS), a species located in the P. putida group. It does not belong to the species P. putida (it only has a dDDH of 25.1 % with P. putida’s type strain). S04 does not belong to any Pseudomonas species described to date (or at least a species for which a genome is publicly available). It only has a 28.2 % dDDH with Pseudomonas frederiksbergensis LMG 19851, the type strain of Pseudomonas frederiksbergensis. You should write Pseudomonas sp. S04. Lastly, S35 is wrongly reported to belong to the species P. fluorescens (the closest type strain is Pseudomonas cyclaminis MAFF 301449, but it only has a dDDH of 44% with this type strain). This strain should be presented as Pseudomonas sp. S35 (but above all not P. fluorescens S35). I understand this is not essential for this manuscript (as there is no genetic and/or genomic content in it), but I encourage your research group to act on these issues, considering it diminishes the quality and scientific soundness of the published manuscripts.

L94 : Add city and country for BEMIS Flexible Packaging (to match other companies listed in the M&M section).

L110 and L112 : Avoid unnecessary abbreviations (O/N). Please write overnight instead.

L118-119 : the in 125’000 is weird. Please replace with , or a space (or nothing ?) depending on the journal style. You can also use scientific notation.

L180-182 : Maybe add one sentence to state the average CFU per leaves after 120h ? ‘’Grow well’’ is a rather imprecise term and the Pseudomonas concentration was higher I expected. It is worthwhile to add a number in my opinion.

L185 : I would like you to add a supplementary table which lists all VOCs detected in this study. I believe it should be interesting for the readers, considering some people might be interested by VOCs produced by the potato leaves (but not so much by those produced by Pseudomonas bacteria.

L188-L189 : ‘’32 VOCs which changed according to the bacterial strain inoculation status of the leaves.’’ I found this part really unclear, and I believe you should rewrite it. I’m not sure what you meant by that. For me, it could mean several things, among which :  i) These 32 VOCs are detected in significantly different amounts between potato leaves inoculated OR NOT with bacteria (suggesting that the remaining VOCs are produced by the plant itself at basal level, and are not produced by the different bacterial strains or by the leaves in reaction to Pseudomonas inoculation) ii) These 32 VOCs are produced in different amounts between the different bacterial strains. Either way, it should be specified.

L258 : One element absent in this discussion is the  amount of anti-Phytophthora VOCs produced and quantified in this experiment. If I make a comparison with phenazine antibiotics produced by plant-beneficial Pseudomonas spp. in the rhizosphere, detecting a little amount of antibiotics in the rhizosphere was an important step (Thomashow 1990), as it meant the bacteria are able to produce antibiotics where it is needed. On the other hand, what was way more interesting was to detect phenazine antibiotics in the rhizosphere of wheat at concentration sufficient to (locally) inhibit pathogen growth (Mavrodi 2012). I don’t know if you can somehow convert QIPA/g in nmol/g and assess the quantity produced . Do you believe these concentrations are sufficient to inhibit P. infestans growth and thwart the infection ? Maybe you could discuss it a little in the discussion (even if it remains speculative).

Author Response

L47-48 : Attributing the Great Hunger of 1845-1849 to Pinfestans in a cause-and-effect relationship is very far from the truth, considering the high number of causes and/or aggravating factors (overpopulation, government failure, mass eviction, extreme poverty etc.). There are a lot of research on this traumatic, complex and polarizing event in the Irish history, and it is important to remember that numerous shipments of grain and other crops were exported from Ireland under military escort at the peak of the famine. I understand why plant pathologists and students like to use this example (one of the best examples of the damage a plant pathogen can cause if left unchecked) and there are not a lot of examples of plant diseases with this level of destruction (the other example that comes to my mind is the thousands of hectares of orchards devastated in the US by the fire blight disease). I would rewrite this sentence to present P. infestans as ‘’A’’ cause of the Irish Famine (and not ‘’THE’’ cause as it is implicitly stated in the current wording) – it could be just by adding ‘’partly’’ before ‘’caused’’.

Thank you for this remark, we have followed your suggestion to avoid presenting a too simplistic and reductionist view of historical events.

L84-85 : Three out of the four bacterial strains used in this study are wrongly identified at the species level in the manuscript. You can use the Type (Strain) Genome Server (https://tygs.dsmz.de/) to check by yourself. Only R47 is correctly identified as a Pchlororaphis strain. R32 belongs to the species Pdonghuensis (digital DDH of 95.5% with the type strain HYS), a species located in the Pputida group. It does not belong to the species Pputida (it only has a dDDH of 25.1 % with Pputida’s type strain). S04 does not belong to any Pseudomonas species described to date (or at least a species for which a genome is publicly available). It only has a 28.2 % dDDH with Pseudomonas frederiksbergensis LMG 19851, the type strain of Pseudomonas frederiksbergensis. You should write Pseudomonas sp. S04. Lastly, S35 is wrongly reported to belong to the species Pfluorescens (the closest type strain is Pseudomonas cyclaminis MAFF 301449, but it only has a dDDH of 44% with this type strain). This strain should be presented as Pseudomonas sp. S35 (but above all not Pfluorescens S35). I understand this is not essential for this manuscript (as there is no genetic and/or genomic content in it), but I encourage your research group to act on these issues, considering it diminishes the quality and scientific soundness of the published manuscripts.

Thank you for making us aware of this point, indeed we used the original classification based only on 16S and which are to a large extent confirmed by our full genome analysis of these strains, but we did not check recent development in the phylogeny of this genus.  We have corrected the names as you suggested.

L94 : Add city and country for BEMIS Flexible Packaging (to match other companies listed in the M&M section).

 Done

L110 and L112 : Avoid unnecessary abbreviations (O/N). Please write overnight instead.

  Done

L118-119 : the  in 125’000 is weird. Please replace with , or a space (or nothing ?) depending on the journal style. You can also use scientific notation.

 Done

L180-182 : Maybe add one sentence to state the average CFU per leaves after 120h ? ‘’Grow well’’ is a rather imprecise term and the Pseudomonas concentration was higher I expected. It is worthwhile to add a number in my opinion.

Indeed, thank you for the suggestion. Number has been added.

L185 : I would like you to add a supplementary table which lists all VOCs detected in this study. I believe it should be interesting for the readers, considering some people might be interested by VOCs produced by the potato leaves (but not so much by those produced by Pseudomonas bacteria.

We agree that this might interest some readers and have therefore added a new supplementary table with all VOCs detected and the respective “abundance” (although quantification is not possible, see below) in all different samples. Thank you for this suggestion.

L188-L189 : ‘’32 VOCs which changed according to the bacterial strain inoculation status of the leaves.’’ I found this part really unclear, and I believe you should rewrite it. I’m not sure what you meant by that. For me, it could mean several things, among which :  i) These 32 VOCs are detected in significantly different amounts between potato leaves inoculated OR NOT with bacteria (suggesting that the remaining VOCs are produced by the plant itself at basal level, and are not produced by the different bacterial strains or by the leaves in reaction to Pseudomonas inoculation) ii) These 32 VOCs are produced in different amounts between the different bacterial strains. Either way, it should be specified.

Thank you, we have reformulated this sentence and hope that it is clearer now. We also added a reference to Table S2 since all data is now presented there.

L258 : One element absent in this discussion is the  amount of anti-Phytophthora VOCs produced and quantified in this experiment. If I make a comparison with phenazine antibiotics produced by plant-beneficial Pseudomonas spp. in the rhizosphere, detecting a little amount of antibiotics in the rhizosphere was an important step (Thomashow 1990), as it meant the bacteria are able to produce antibiotics where it is needed. On the other hand, what was way more interesting was to detect phenazine antibiotics in the rhizosphere of wheat at concentration sufficient to (locally) inhibit pathogen growth (Mavrodi 2012). I don’t know if you can somehow convert QIPA/g in nmol/g and assess the quantity produced . Do you believe these concentrations are sufficient to inhibit Pinfestans growth and thwart the infection ? Maybe you could discuss it a little in the discussion (even if it remains speculative).

We unfortunately cannot estimate the absolute quantity, as the collection method used (SPME) as well as the absence of an internal standard does not allow to convert the peak areas (QIPA) to molar concentrations. We are still a very long way from asserting that the protective effect of the strains would be due to volatiles. It will likely depend on the leaf colonization pattern (are bacteria producing biofilms at specific microsites as described in the literature and producing locally high concentrations at these hotspots of activity?), on the resources available (different cultivars might sustain different bacterial abundance on their leaves) and on the specific activity of the compounds. Some need micromolar concentrations to be active, other inhibit zoospore germination in the femtomolar range (see our scientific report paper on MMTS, doi: 10.1038/s41598-019-55218-3). We agree that this is an essential point though, and added a sentence in the discussion to make readers aware of it, until we can bring more quantitative evidence in future work.

Reviewer 2 Report

The present study investigated the emission of bioactive volatiles by phyllosphere bacteria Pseudomonas. The research work presents an interesting topic and is well organized. The methodological part is also well described.

I have provided some comments and suggestions, which may help authors improve their Manuscript. 

Comments

Line 70. It would be very helpful for readers if you describe phyllosphere bacteria, surface colonizing, or endophytes? If plants were inoculated with these bacteria, how was inoculation performed?

Line 175: “Pseudomonas strains (R32, R47, S04 and S35” why four strains were chosen for plant inoculation. Are they differ in their PGP traits?

Line 178: Are these four strains exhibit antagonistic activity against P. infestans?, e.g. in previous studies?

Figure 2, It is difficult to read; please make the text readable or change the format of the figure.

It would be really useful to describe four selected Pseudomonas bacteria for the plant inoculation. Are they differ in their physiological properties or PGP traits?

References: Please add more recent reports on VOC produced by phyllosphere bacteria

Author Response

Line 70. It would be very helpful for readers if you describe phyllosphere bacteria, surface colonizing, or endophytes? If plants were inoculated with these bacteria, how was inoculation performed?

Thank you for this comment. The inoculation is described in LL118-119. The original isolation of the strains is reported in the cited paper (Hunziker et al.) and there, we mention that endopyhtes and epiphytes could not be differentiated, as leaves were ground and serially diluted before plating to recover isolates.

Line 175: “Pseudomonas strains (R32, R47, S04 and S35” why four strains were chosen for plant inoculation. Are they differ in their PGP traits?

Thank you for this comment. We have added the reasons for choosing these strains in the material and method section (LL86-87).

Line 178: Are these four strains exhibit antagonistic activity against P. infestans?, e.g. in previous studies?

Thank you for this comment. Please see above (LL86-87). Their activity on P. infestans differed, as described in earlier papers from our group (referenced in the material and method section, when we introduce the strains).

Figure 2, It is difficult to read; please make the text readable or change the format of the figure.

Thank you for this comment. We have worked on the figure to make more readable and hope that it will be clearer now.

It would be really useful to describe four selected Pseudomonas bacteria for the plant inoculation. Are they differ in their physiological properties or PGP traits?

Thank you for this comment. Since their activity has been described in great many details in earlier work (see e.g. 10.3389/fmicb.2020.00857), we believe that it would be out of scope of the present study to restate it here, as the main point of this paper is to show that VOCs are emitted on leaf material, but not specifically to link this activity with the strains’ activities in vitro. The main interesting point is that the strain which emitted most volatiles was not active in in vitro experiments but protected well in planta, and this information is already in the paper (LL318-319).

References: Please add more recent reports on VOC produced by phyllosphere bacteria

We are not aware of more recent papers on VOCs emitted ON LEAVES by phyllosphere bacteria. Would the reviewer kindly suggest which papers we have been missing?

Reviewer 3 Report

This manuscript reportede that bacterial VOCs have potential to inhibit the phytopathogen, P. infestans. It is well known that bacterial VOC inhibits the phytopathogen, but this manuscript demonstrated that bacteria emitted VOC on the phytosphere. This is a novel finding and have a value to be published. However, it is difficult for readers to understand the findings indicated data. If there are some supplementary materials such as photographs, readers will be able to understand this research. And it is well known that bacterial hydrogen cyanide produced by many pseudomonads inhibit both of microorganisms and plants, please challenge to reveal the effect of HCN in further study.

Author Response

Thank you for these comments. We were not able to identify which particular steps were not sufficiently described to allow to understand the findings. Perhaps our newly added supplementary table 2 showing the data in a more comprehensive manner (all detected VOCs in all treatments) will be helpful in this regard, at least we hope so. Regarding HCN emission, indeed it is an important component of the volatiles emitted. From the four strains we selected, two were able to produce HCN in vitro but it is not clear whether they would emit this respiratory toxin on plant leaves (HCN cannot be detected with GC/MS due to its low mass). We rather think that is would not be emitted, as our previous work showed that HCN mutants in these two cyanogenic strains did not lead to loss of protective ability in planta, although strong losses of inhibitory potential was observed in vivo (please see : 10.3390/microorganisms8081144).